# R2G Loss Accelerates Grokking in Transformer Models

## Abstract

Transformers, whose wide success makes them central to grokking research, rely on inter-token interactions within attention layers. Through experiments on attention mechanisms, we find the distributional differences in the representation space of three phases (memorization, semi-grokking and grokking). We observe that during the grokking phase, the model develops a structural separation of tokens and learns both the dual characteristics (symbolic and numerical) of input data. Based on these findings, we propose R2G (Repel to Grokking) Loss: a simple method to accelerate grokking with finite data, which can foster higher-level generalization. Empirical studies on different arithmetic tasks demonstrate that R2G Loss effectively modulates training dynamics, leading to significant performance improvement under identical input. Our method is validated on different arithmetic tasks and a non-arithmetic task, in both of which we achieve an improvement in the model's grokking. In modular arithmetic tasks, we are even able to achieve grokking in situations where training has previously failed. Our work offers a novel mechanistic understanding of grokking, along with a simple and versatile tool to accelerate the grokking process in transformer models. These findings also have the potential to inspire effective enhancement of the model's generalization capability across a broader range of scenarios.

## 1 Introduction

While artificial neural networks have demonstrated impressive performance across many tasks, understanding their internal mechanisms remains a major challenge. The study of "grokking" — a recently discovered phenomenon where neural networks abruptly transit from memorizing training data to achieving perfect generalization after prolonged training (Power et al., 2022) — provides a crucial perspective for understanding the learning process of these models. This phenomenon, initially observed in transformers performing modular arithmetic, has since been replicated across diverse domains, including group operations (Chughtai et al., 2023), sparse-parity tasks (Barak et al., 2022), image classification (Liu et al., 2023), and structural grokking on language processing tasks (Murty et al., 2023). Understanding this transition from memorization to deeper pattern comprehension is vital, there are two main reasons: (1) Theoretically, it contributes to understanding the internal mechanisms of model generalization; (2) Practically, the widespread phenomenon provides an insight to enhance model capabilities.

The discovery of grokking raises a fundamental question: how do models acquire generalization capabilities in these tasks? Some previous works attribute grokking to learning well-structured representations matching task-specific requirements (Power et al., 2022) (Liu et al., 2022). While some researchers have interpreted internal model mechanisms through the "pizza algorithm" and "clock algorithm" (Zhong et al., 2023) (Nanda et al., 2023). Varma argued that the apparent suddenness of grokking stems from a shift in dominance between memorization and generalization algorithms, where generalization emerges only after the model has fully suppressed its memorization mechanisms (Varma et al., 2023). Huang et al. find that the model's performance can be divided into four phases according to model hidden size and train data size (Huang et al., 2024). And under a fixed hidden size, progressively increasing train data size improves the generalization level of models, leading to a phase evolution from memorization to semi-grokking to grokking.

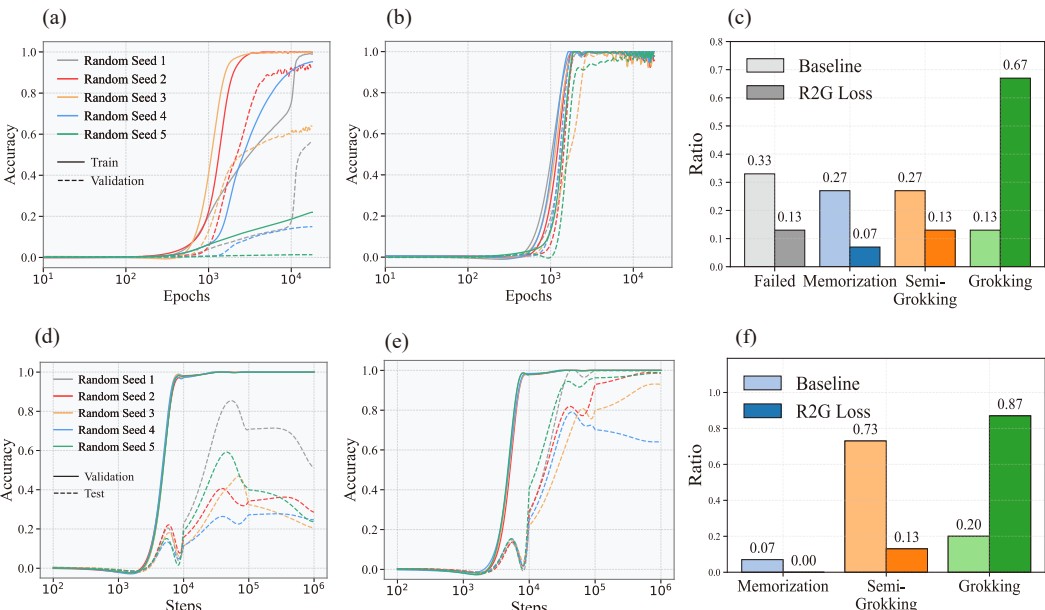

Figure 1: The R2G Loss accelerates grokking. (a-c) The modular $P = 500$ arithmetic task. (d-f) The tense-inflection task. Under identical input data, the (a,d) original training curves, and (b,e) R2G Loss training curves after smoothing show that different original training states are successfully promoted to achieve grokking after applying R2G Loss method. (c,f) The improvement brought by R2G Loss over baseline across four distinct tasks: Failed, Memorization, Semi-grokking, and Grokking. Notably, R2G Loss demonstrates significant gains in the Grokking category.

Many studies are dedicated not only to understanding grokking but also to eliminating grokking by reducing the time delay between memorization and generalization. Techniques include constraining weight norms within a spherical radius (Liu et al., 2023), initializing with pre-grokked weights (Furuta et al., 2024), amplifying specific gradient components (Lee et al., 2024), and applying lottery ticket masks (Minegishi et al., 2025). Notably, GrokTransfer achieves delay reduction without pre-trained models or additional data (Xu et al., 2025).

However, previous studies have not focused on how internal representations drive the transition from memorization to semi-grokking to grokking phases, which is crucial for extracting features from data and completing tasks. Understanding the internal mechanisms of performance improvement provides valuable insights for designing effective methods to accelerate grokking. Given that grokking research primarily focuses on transformers, where attention is a central part, our work investigates the phase evolution process on the attention layer. By examining internal dynamics across phases, we discover distinct distributional shifts in the representation space, revealing the model's distinct understanding of data's dual characteristics and proposing a novel structural perspective on understanding grokking. Based on the architectural patterns discovered, we design R2G (Repel-to-Grokking) Loss to accelerate the emergence of higher-level generalization, achieving superior performance with fewer data and computational costs. Specifically, we study several arithmetic tasks and a non-arithmetic task. In the grokking mechanism exploration part, we study on the $P = 500$ modular-addition benchmark, where a model processes inputs $\{a, +, b, \%, P, =\}$, and $a, b \in (0, 1, 2, \ldots, P - 1)$, to predict $c \equiv a + b(modP)$. In the grokking acceleration part, we validate the R2G Loss method on mathematical tasks and a language processing task. Our contributions are as follows:

1. **Investigation of Performance Improvement Process**: Focus on memorization, semi-grokking, and grokking phases, we conduct model mechanism studies through tracking structural evolution of embeddings in the attention layer by employing dimensionality reduction, visualization techniques, and statistical analysis.

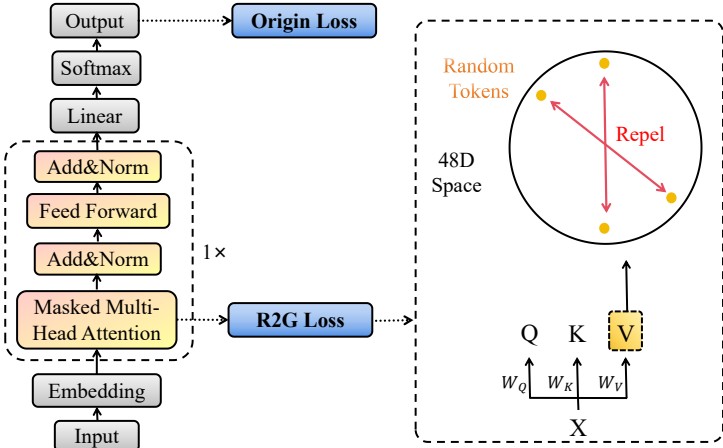

Figure 2: Diagram of R2G Loss in 1-layer decoder-only transformer: a simple method based on embedding structure in the attention layer, it accelerates grokking by enforcing structural separation of random $V$ embedding vectors in 48D hidden space.

2. **Discovery of Grokking Mechanisms**: Inputs initially represent meaningless symbols to models. In experiments on the arithmetic dataset, we observe that the model exhibits different understandings of the dual characteristics of data across different phases:

   - **Symbolic (▲, ■, •, ♦):** In the memorization phase, the model can distinguish different symbols but cannot determine the magnitude of numbers or perform calculations.
   - **Numerical (**$1, 2, 3, 4$**):** In the grokking phase, the model can to some extent understand the difference of numbers.

3. **Design of R2G Loss**: We establish correlations between the microscopic structure of embeddings and macroscopic generalization, based on which we design the ***R2G (Repel-to-Grokking) Loss*** to promote the model's emergent capabilities. This method does not require prior information, which accelerates phase transition by enforcing geometric separation in embedding space (Figure 1), and its effectiveness is validated across different tasks. The four phases are defined according to the models' train-test performance gaps:

   - **Failed**: Even after a sufficiently long training period (e.g., $10^4$ epochs), the training accuracy still does not reach a perfect level.
   - **Memorization**: Memorization dominates, with high training accuracy and low validation accuracy.
   - **Semi-Grokking**: Comparable efficiency allows partial generalization post-memorization.
   - **Grokking**: Generalization dominates, yielding near-perfect validation accuracy.

## 2 RELATED WORK

**Transformer.** Since its introduction in 2017, the transformer model (Vaswani et al., 2017) has emerged as the core architecture in natural language processing (NLP) (Chapagain & Rus, 2025), computer vision (CV) (Yan et al., 2024), and multimodal learning 2025(Wang et al., 2025). Its primary innovation lies in the multi-head self-attention mechanism, which enables parallel computation of global dependencies among tokens. This mechanism effectively addresses the challenges of modeling long-range dependencies in recurrent neural networks (RNNs) and overcomes the local receptive field limitations of convolutional neural networks (CNNs). The original transformer architecture is composed of encoder and decoder modules. And many variants have been developed, which can be categorized into several types according to their architecture and applications. For example, BERT (Devlin et al., 2019) employs an encoder-only architecture, utilizing deep bidirectional self-attention for contextual semantic modeling, which significantly enhances performance on text-understanding NLP tasks. In contrast, GPT (Radford et al., 2019) adopts a decoder-only

architecture, leveraging masked self-attention for autoregressive text generation, and demonstrates outstanding performance in language modeling and generative tasks.

**Loss Function Design.** The loss function serves as a guide for the optimization of model, and its design is critical to effective training. Researchers have proposed various loss functions tailored to specific tasks. For instance, regression losses such as Mean Squared Error (MSE) and Mean Absolute Error (MAE) are widely used for predicting continuous values. Classification loss fuctions measure the discrepancy between predicted results and ground-truth labels, with common examples including Cross Entropy Loss and Dice Loss. Optimizing loss functions remains a key direction for improving the training process. For example, Wasserstein GAN (Arjovsky et al., 2017) fundamentally redefines the GAN training mechanism by incorporating the Wasserstein distance as a distributional measure. CIDER (Ming et al., 2022) jointly optimizes two loss functions to effectively improve out-of-distribution (OOD) detection performance. More recently, He et al. introduce Dispersive Loss for diffusion models to enhance sampling diversity and quality (Wang & He, 2025).

## 3 METHOD

### 3.1 MODEL

In our work, we train a 1-layer decoder-only transformer model with 4 attention heads. During training, a cross-entropy loss function ForCausalLMLoss with the AdamW (Loshchilov & Hutter, 2019) optimizer, using a learning rate of 0.001 and a weight decay of $\gamma = 0.01$. The model has width $d_{model} = 48$ with GeLU activations. The train data size ranges from 8000 to 40000 to cover the three stages of memorization, semi-grokking, and grokking. Validation dataset size is fixed as 2000 (Appendix A.1).

### 3.2 QKV PATTERN ANALYSIS

The Query ($Q$), Key ($K$), and Value ($V$) vectors form the fundamental elements of the attention computation. We analyze the $Q, K, V$ patterns using Principal Component Analysis (PCA) dimensionality reduction and Euclidean distance measurements. Results demonstrate that model grokking strongly correlates with latent-space representations of $V$ vectors.

Consequently, by visualizing microscopic representational dynamics of $Q, K, V$ vectors, we aim to precisely describe the inner processes governing the grokking phenomenon within the transformer's attention mechanism. Building on these observations, we propose a simple method to accelerate grokking.

### 3.3 R2G LOSS

We design a new loss function that provides an extra force by injecting symbolic geometric structure into the optimization objective. The main idea behind our approach is to enlarge the distance between randomly sampled $V$ vector pairs in the original 48D embedding space, the diagram is shown in Figure 2. Therefore, we term this loss the *Repel to Grokking (R2G) Loss*, defined as follows:

$$\mathcal{L}_{R2G} = distance(H_{\text{embedding}} - H'_{\text{embedding}}), \tag{1}$$

$$\mathcal{L} = \mathcal{L}_{\text{orign}} + \alpha \cdot \mathcal{L}_{R2G}, \tag{2}$$

where the distance is measured using the Euclidean distance, $H_{\text{embedding}}$ specifically denotes the embedding of the value vectors $V$ in the attention layer of transformer, and $H'_{\text{embedding}}$ represents its randomly shuffled counterpart, enabling the calculation of the distance between random couples of embeddings. $\alpha$ is a tunable hyperparameter to control the weight of R2G Loss. When $\alpha = 0$, it completely degenerates into the original model. $\alpha$ needs to be tuned for different tasks to achieve the best performance, the settings and selection criteria are detailed in Appendix B.2.

Our method does not use any additional data or introduce extra modules. The R2G Loss relies solely on the intermediate representations from the attention layers and does not require introducing any other learnable parameters. The specific form of the R2G Loss is defined in Algorithm 1. Such simplicity will enable it to be adapted to various tasks.

---

**Algorithm 1** R2G Loss

---

1: **def R2G_Loss**$(V_a, V_b, V_c)$:
2:    *Input:* $V_a, V_b, V_c \in \mathbb{R}^{B \times D}$, *Output:* $L_{\mathrm{R2G}} \in \mathbb{R}$
3:    # $B$ = batch size, $D$ = embedding dim, $V_i$ denotes the value tensor of operand $i \in a, b, c$.
4:    $perm \leftarrow \mathrm{RandomPermutation}(B)$
5:    $V_i' \leftarrow V_i[perm, :], i \in \{a, b, c\}$
6:    $L_{\mathrm{R2G}} \leftarrow \sum_{i \in \{a,b,c\}} \|V_i - V_i'\|_2$
7: **return** $L_{\mathrm{R2G}}$

---

### 3.4 TASK

Since the transition from memorization to semi-grokking to grokking is more easily observed in binary mathematical operations, and the model training of mathematical operations is highly efficient, we first conduct the main experiments on mathematical operations and use modular addition as a representative example for clarity (Power et al., 2022):

$$(a + b) \bmod P = c, \tag{3}$$

$a, b \in (0, \dots, P - 1)$, $P \in \mathbb{N}$. Our experiments encompass cases where $P$ is both prime and composite. $P = 500$ was chosen for presentation in the main text, because a larger modulus provides a broader scope for investigation, where the exhibited patterns demonstrate greater generality. In later sections, we will also present experimental results for other moduli $P$ and different operations. These tasks are viewed as classification tasks, where the label number is $P$. We split training and validation subsets by employing distinct $(a, b)$ tuple combinations. For simplicity, we directly use validation performance to distinguish memorization, semi-grokking, and grokking. More comprehensive data details are available in Appendix A.2.

Meanwhile, to verify the generality of our proposed R2G Loss, we conduct experiments on a non-mathematical task — the tense-inflection task, a language processing task that leverages English subject–verb agreement to reveal a model's syntactic generalization ability (Linzen et al., 2016). Since the statements have a hierarchical structure, structural grokking will occur in this task (Murty et al., 2023). The model receives an English sentence in the past tense and a tense marker (PAST or PRESENT) as input, and then output a sentence whose tense is converted according to the tense marker. When the sentence is required to be present tense, the model must determine whether each verb should take the singular or plural form. There are two rules for making this determination (McCoy et al., 2020):

(1) AGREE-RECENT: Each verb should agree with the linearly most recent noun.
(2) AGREE-SUBJECT: Each verb should agree with its hierarchically determined subject.

Although these rules yield the same predictions for sentences with linear structure, they diverge on sentences with complex hierarchical structure, such as the following sentence (a), where the correct AGREE-SUBJECT prediction is (b), while the wrong AGREE-RECENT prediction is (c):



(a) my zebra by the yaks swam . PRESENT
(b) my zebra by the yaks swims .
(c) my zebra by the yaks swim .



We evaluate structural grokking using the out-of-distribution test set. The specific model configuration for this task is detailed in the Appendix B.1.

## 4 GROKKING WITH REPEL TOKENS IN HIDDEN SPACE

In self-attention mechanisms, the query ($Q$), key ($K$), and value ($V$) vectors encode distinct functional roles. And attention scores emerge from interactions between these vectors. To investigate finer-grained dynamics, we analyze $Q$, $K$, and $V$ vectors independently before their combination.

- **Query** ($Q$): Query vectors encode the information demand of the current token. Formally, $Q = XW_Q$ represents "what the token seeks to know" from the context.

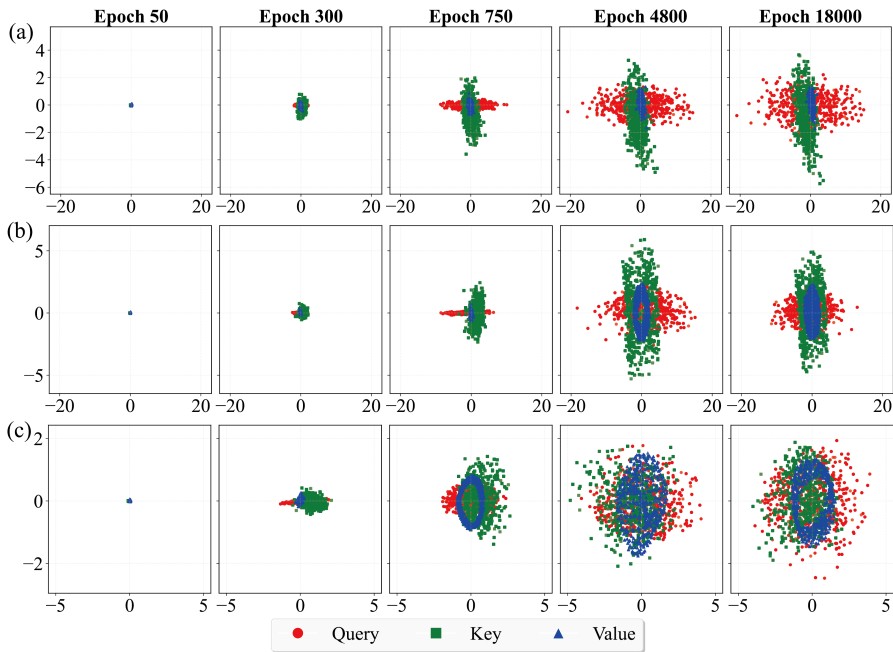

Figure 3: Embedding of $Q, K, V$ vectors of 2000 validation data in a two-dimensional space after PCA, and five representative epochs are selected for each case. (a) memorization($N = 8000$). (b) semi-grokking($N = 25975$). (c) grokking($N = 40000$). The figure exclusively illustrates the results corresponding to operand $a$, while results for $b$ and $c$ are detailed in Appendix C.1.

- **Key** ($K$): Key vectors define retrievable features for each token. Each $K = XW_K$ acts as an addressable descriptor answering "what this token can provide". Attention weights can be computed by alignment scores between $Q$ and $K$ vectors:

$$\mathrm{softmax}\left(\frac{QK^T}{\sqrt{d_{model}}}\right). \tag{4}$$

- **Value** ($V$): While Q and K vectors govern attention allocation, value vectors $V = XW_V$ contain actual content for propagation: When attention weights determine "where to look", $V$ vectors define "what information to transmit". Visualizing $V$ vectors provides direct access to the model's learned representations.

Specifically, we apply PCA to reduce the dimension of $Q, K, V$ vectors, which reveal latent structural patterns obscured in high-dimensional space. PCA is preferred as it preserves geometrically meaningful variance, unlike t-SNE whose nonlinear projections mask discriminative features. Evolution analysis of these geometric configurations shows internal mechanistic transitions during grokking.

From Figure 3, we observe distinct behaviors of the $Q, K$ and $V$ vectors. Compared to the memorization phase, the $Q, K, V$ vectors are more widely dispersed in high-dimensional space during grokking. This dispersion indicates that the model makes fuller use of its representational capacity by assigning different numbers to separate regions. During memorization, $Q, K$ vectors form approximately orthogonal structures in the 2D projection: this orthogonality reduces feature similarity, enabling sharper attention weighting. Notably, when grokking, $V$ vectors organize into circular patterns. Since circles in 2D space correspond to hyperspheres in higher dimensions where points maximize pairwise distances, this motivates our investigation: *Does model grokking correlate with distance between embedding vectors?*

To investigate this problem, we visualize the value vectors of operand $a$, for token 1~10 (11~20, 21~30, etc., follow the same rule, the results are detailed in Appendix C.2). Figure 4 reveals that during memorization and semi-grokking, embeddings are distributed randomly, whereas in the grokking

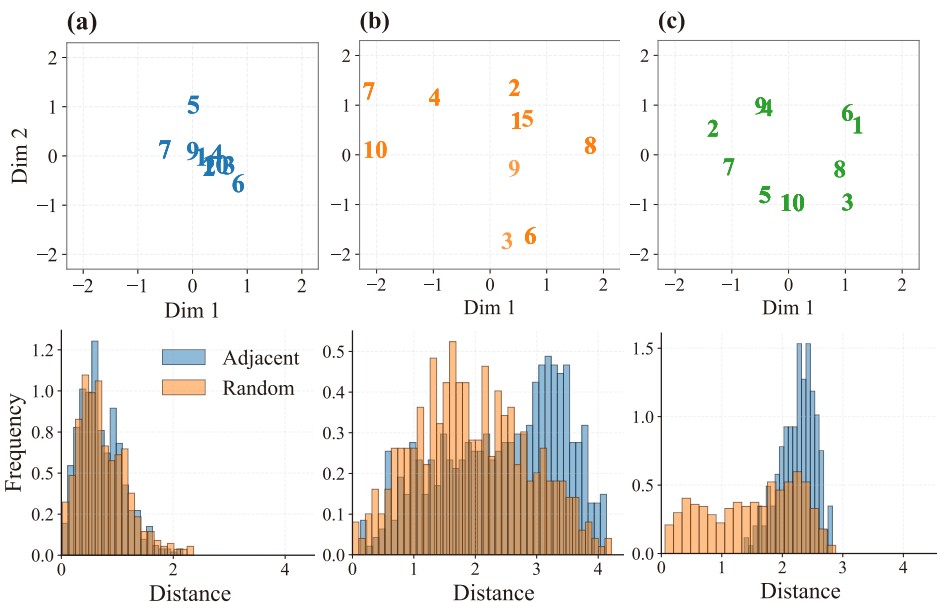

Figure 4: Row1: The embedding position of token 1~10 of operand $a$ (same for the results of operand $b$ and $c$ in Appendix C.3); Row2: Histogram of Euclidean distances between 48-dimensional embeddings on validation data (blue) and random pairs (red). (a) memorization($N = 8000$) (b) semi-grokking($N = 25975$) (c) grokking ($N = 40000$)

phase, they form a circular topology with maximally separated adjacent digits. Histograms in Figure 4 validates this observation through embedding distance in a 48-dimensional space:

- **Memorization**: Adjacent-digit distances (red) match random digit pairs (blue), confirming random embeddings;
- **Semi-grokking**: Adjacent-digit distribution diverges from random ones;
- **Grokking**: Adjacent digits exhibit systematically larger distances than random ones.

From the perspective of a model devoid of prior knowledge, input data exhibits dual characteristics: numerical magnitude and symbolic identity. Our findings demonstrate that during grokking, the model comprehends numerical adjacency and encodes an adjacency repulsion principle: numerically adjacent digits become maximally separated in embedding space. This indicates model's partial acquisition of numerical properties, prompting our investigation: *To what extent can the model achieve semantic understanding of numerical attributes?*

We further visualize the relationship between numerical difference of tokens (1-499) and 48D embedding distance across memorization, semi-grokking, and grokking phases in Figure 5(a)-(c) through scatter plots and histograms. Key observations include:

- **Memorization**: Embedding distances remain random regardless of numerical differences;
- **Semi-grokking**: Partial discrimination of numerical differences but more dispersed distance distributions;
- **Grokking**: Partial discrimination of numerical differences with concentrated distance distributions.

These findings demonstrate improved representation of input numerical invariants during the grokking phase. In Figure5(d), we compute position embedding divergence by summing the Wasserstein distance between data of operands a&b and operands a&c. The observed declining divergence from memorization to grokking demonstrates that models attenuate positional encoding influence when achieving full generalization, preserving only core properties of data.

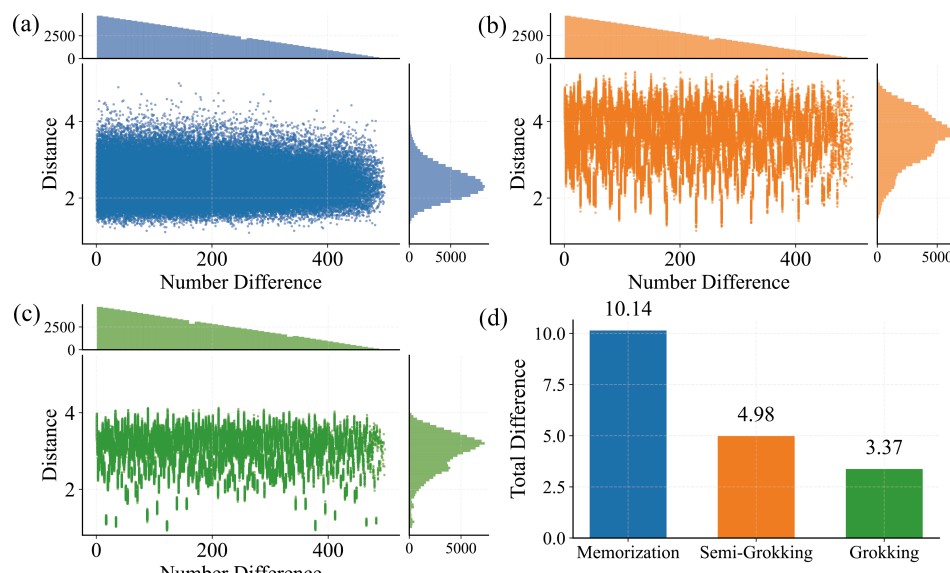

Figure 5: Scatter plot of the original numerical difference (1~499) and the 48-dimensional spatial distance. (a) Memorization($N = 8000$), (b) Semi-grokking ($N = 25975$). (c) Grokking ($N = 40000$). (d) Difference of embedding distribution on operands: $a, b$ and $c$ in three phases.

## 5 R2G LOSS ACCELERATES GROKKING

Capability emergence is central to deep learning, with grokking as a key example, making its promotion a challenge. Motivated by our empirical findings on embedding patterns, we explore whether such structures can be exploited to promote grokking. The design of the new loss function could stem from either the symbolic or numerical nature of the data. However, incorporating numerical priors might confine applicability to datasets. To ensure generality, we focus solely on symbolic priors. We inject symbolic geometric structure into the optimization objective by enlarging the Euclidean distance between randomly sampled $V$ vector pairs in the original embedding space. This is achieved by designing the R2G Loss, whose formula is given by equation 1.

For modular arithmetic tasks, R2G Loss substantially accelerates the grokking process. The main experiment on modular addition with $P = 500$ is shown in Figure 1(a,b,c). Under identical training data of 26,650 samples, introducing R2G Loss dramatically increases the proportion of runs reaching full grokking from 13% to 67%, while sharply reducing stagnation in the semi-grokking, memorization and failed regimes (from 27% to 13%, from 27% to 7% and from 33% to 13%, respectively). Notably, we observe that R2G Loss is capable of not only advancing semi-grokking models to grokking but also rescuing runs that fail to train, elevating a significant portion of 'Faied' cases into the grokking phase. This robust shift in phase transition highlights symbol-driven geometric regularization as an effective, generalizable grokking catalyst in deep neural networks.

We evaluate the capability of R2G Loss on additional prime modular tasks $P = 499$ and $P = 113$. In both settings, R2G Loss consistently and significantly accelerates the grokking process. As shown in Figure 6(a), for $P = 499$ with a training set of 26,600 samples, R2G Loss increases the proportion of grokking runs from 20% to 65%, while reducing the failed from 7% to 0%, memorization phase from 40% to 20% and the semi-grokking phase from 33% to 13%. Similarly, for $P = 113$ with 2,650 training samples, R2G Loss raises the grokking rate from 36% to 60%, and decreases the memorization and semi-grokking phases from 27% to 20% and from 40% to 20%, respectively (6(b)). These results demonstrate that R2G Loss effectively drives models from memorization to high generalization and serves as a general catalyst for grokking, consistently enhancing performance across different modular tasks.

We further evaluate R2G Loss on modular multiplication tasks ($a \times b \% p = c$) to assess its generality. As reported in Figure 6(c), R2G Loss improves generalization performance in this setting. With a

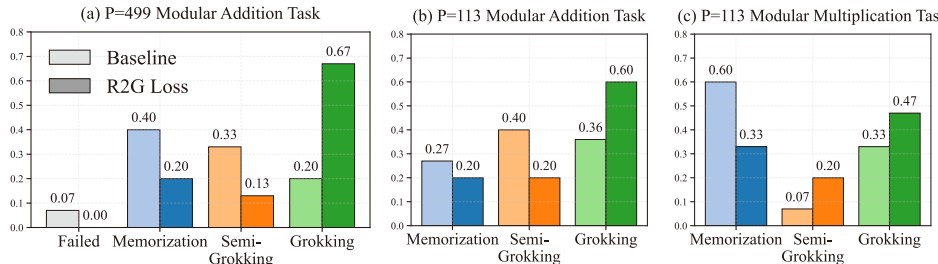

Figure 6: The changes brought by R2G Loss over the baseline of three phases. (a)The result for the $P = 499$ modular addition task. (b)The result for the $P = 113$ modular addition task. (c)The result for the $P = 113$ modular multiplication task.

dataset size of 2,650 and $P = 113$, the grokking rate increases from 33% to 47%. Meanwhile, the proportion of runs stuck in memorization falls from 60% to 33%.

Finally, to ensure general applicability and broader impact, we conduct the same experiments on the tense-inflection task. Further experimental details are provided in Appendix D. The example training curves for the original loss and the R2G loss are shown in Figure 1(d,e). After applying the R2G loss, the grokking rate increases from 20% to 87% with a dataset size of 10,000. Meanwhile, the proportion of semi-grokking decreases from 73% to 13% (Figure 1(f)). To further refine our results, we present the proportions of each phase transition of 5 tasks in the Appendix D.3. These results highlight the effectiveness of R2G Loss in shifting models out of low generalization, demonstrating its role as a powerful and general catalyst for grokking in diverse arithmetic tasks.

The effectiveness of R2G Loss may arise from its ability to promote token representation disentanglement. Tasks usually require precise symbolic encoding, our loss function actively guides the network to enhance inter-token separability, which in turn amplifies discriminative information for token identity and induces the emergence of intrinsic structures within the model's latent space.

## 6 CONCLUSION

Research on the grokking phenomenon offers a vital perspective for understanding the internal mechanisms in machine learning models. Notably, it's a common phenomenon for models to not fully grasp all the mechanisms of data, which poses a critical question: how can models learn more mechanisms and improve their generalization capability? To investigate this, we analyze the internal processes of transformer models performing modular arithmetic tasks on the critical attention layer. Employing principal component analysis (PCA), visualization, and statistical methods, we examine the value vector embeddings throughout the memorization, semi-grokking, and grokking phases. Our study on the arithmetic task reveals that dual data characteristics (symbolic and numerical) govern the cognitive evolution of model: During memorization, the model primarily acquires symbolic characteristics, while in the grokking phase, it also develops an understanding of numerical characteristics.

Inspired by these findings, we design a loss function to accelerate the grokking process. In real-world tasks, symbolic characteristics are widely exhibited in data. To develop a universally applicable tool for diverse tasks, we design the R2G Loss based on symbolic aspects — a geometrically principled method that actively promotes model competence by enforcing structural separation between embeddings of value vectors in the attention layer. Validation across arithmetic and non-arithmetic tasks demonstrates the versatility of R2G Loss, which effectively alters training dynamics and enables significant improvements in model performance under identical inputs.

Although R2G Loss demonstrates impressive performance in arithmetic and non-arithmetic tasks, an important direction for future research is extending its capabilities to more complex and real-world problems. Moreover, the potential of R2G Loss to facilitate generalization in non-grokking scenarios also deserves further investigation. Our study offers valuable insights and academic contributions to understanding emerging capabilities and improving model performance.

## 7 REPRODUCIBILITY STATEMENT

This work strictly follows the principles of reproducibility, with detailed descriptions of all main experimental methods and parameter settings. The paper provides conceptual outlines and pseudocode for the proposed AI methods, clearly distinguishes between objective results and subjective statements, and offers pedagogical references for background knowledge.

Method and implementation details are provided in Section 3.2, 3.3, 3.4, and Appendix D.2. Experiment settings and details are described in Section 3.1, Appendix B.1, B.2 and D.1. All experimental details and key parameters are fully disclosed, ensuring that any researcher with a relevant background can independently reproduce the reported results by following the instructions and resources provided in the paper. All code and data for these experiments is available at `https://anonymous.4open.science/r/R2G_Loss_accelerates_grokking_code`

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

# Appendix

## A  DATASET

### A.1  DATA USED FOR GROKKING/SEMI-GROKKING/MEMORIZATION

The main experiments in this paper are conducted on the following three datasets:

- Memorizationtraining data size $N = 8000$ (Figure7(a))

- Semi-grokking: training data size $N = 25975$ (Figure7(b))

- Grokking: training data size $N = 40000$ (Figure7(c))

For each operation, we construct a dataset of equations with the form $\langle a\rangle\langle op\rangle\langle b\rangle\langle\%\rangle\langle P\rangle\langle=\rangle\langle c\rangle$, where $\langle x\rangle$ stands for the token corresponding to element x.

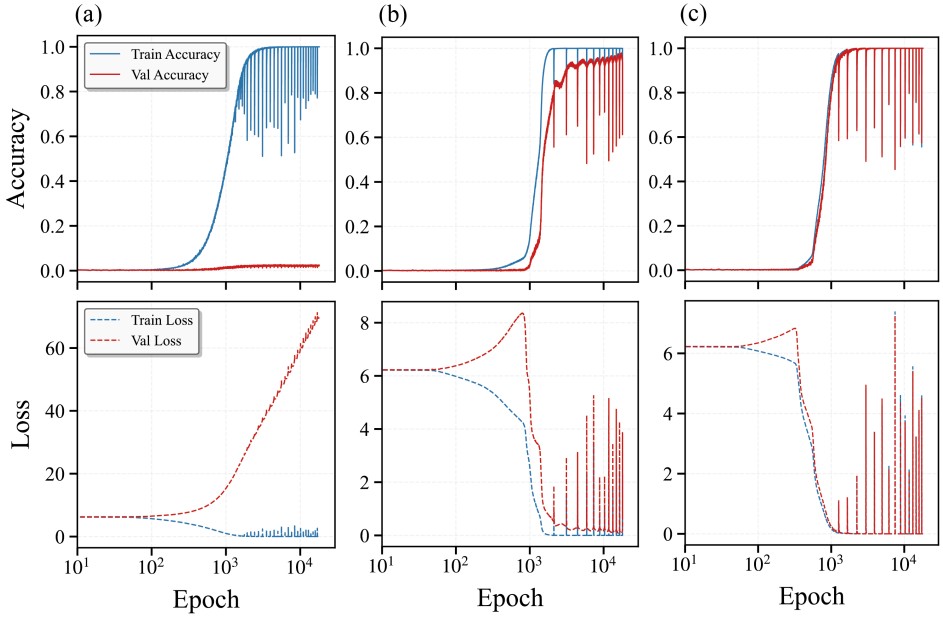

Figure 7: Accuracy and loss curves for three phases in the main article. (a) Memorization. (b)Semi-grokking. (c) Grokking.

### A.2  DATA GENERATION

The following are the operations that we have tried in our work:

- $(a + b) \bmod P = c\,(P = 500)$ for $0 \le a, b < P$

- $(a + b) \bmod P = c\,(P = 499)$ for $0 \le a, b < P$

- $(a + b) \bmod P = c\,(P = 113)$ for $0 \le a, b < P$

- $(a \times b) \bmod P = c\,(P = 113)$ for $0 \le a, b < P$

For each training run, we first enumerate all possible modular arithmetic equations in the range $0 \le a, b < P$. From these, we randomly sample $\mathrm{train\_size}$ data as the training set and 2,000 as the validation set.

## B  MODEL AND PARAMETER SETTINGS

### B.1  MODEL FOR TENSE-INFLECTION TASK

In the tense-inflection task, we use an encoder-only Transformer model with the attention mask, using the following hyperparameters:

- Number of encoder layers = 4

- Number of attention heads = 8

- Hidden dimensionality = 512

- Batch size = 8.

- Loss Function: Cross Entropy Loss

- Optimizer: AdamW ($\beta_1 = 0.9$, $\beta_2 = 0.999$, $\epsilon = 1 \times 10^{-4}$)

### B.2  SETTINGS OF R2G LOSS STRENGTH PARAMETER $\alpha$

Due to differences in tasks and parameter settings, users may have varying preferences for the choice of $\alpha$ in order to control the strength of R2G Loss, which to encourage token separation and achieve optimal performance.

We report the best-performing $\alpha$ from all those we tried in Table 1, but due to experimental resource constraints, even better values of $\alpha$ may remain to be discovered. In the primary experiment (modular addition with $P = 500$), we set $\alpha = 0.3$. For the variants with $P = 499$ and $P = 113$, we used $\alpha = 0.27$ and $\alpha = 0.08$, respectively. In the modular multiplication experiment with $P = 113$ we used $\alpha = 0.01$, and in the tense-inflection task, we set $\alpha = 0.001$.

| Task | Setting | $\alpha$ |
|---|---|---|
| Modular addition | $P = 500$ | 0.3 |
| Modular addition | $P = 499$ | 0.27 |
| Modular addition | $P = 113$ | 0.08 |
| Modular multiplication | $P = 113$ | 0.01 |
| Tense-Inflection Task | – | 0.001 |

Table 1: $\alpha$ settings for different tasks.

In the current experiments, we observe that the optimal value of $\alpha$ may relate to the scale of loss, tasks, and the number of tokens. There is a tendency that the more tokens need to be separated, the larger $\alpha$ is recommended. To simplify the use of our method, we hope to explore approaches to automate the selection of the parameter $\alpha$ in the future.

## C  VISUALIZATION

### C.1  VISUALIZATION OF $Q, K, V$ EMBEDDING VECTORS

Figures8 and 9 present the 2D PCA embeddings of the Query ($Q$), Key ($K$), and Value ($V$) vectors for operands $b$ and $c$, respectively. The dataset we use are 2,000 validation samples, same as employed for operand $a$ in the main article. The results demonstrate that the embeddings for operands $a$, $b$, and $c$ are nearly identical. This indicates that the model largely disregards positional embedding differences of data.

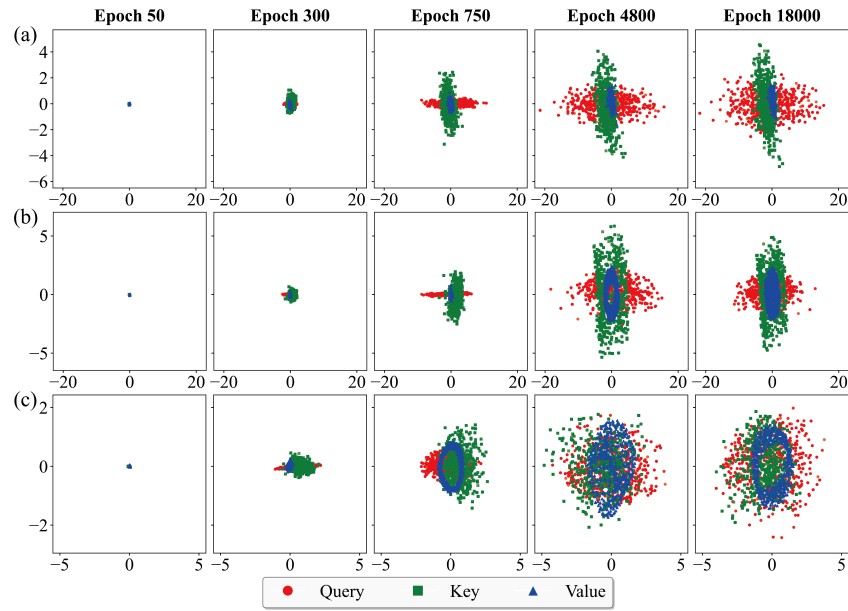

Figure 8: Embedding of query, key, and value vector in 2D space after PCA, and five representative epochs are selected for each case. (a) memorization. (b) semi-grokking. (c) grokking. The figure exclusively illustrates the results corresponding to operand $b$.

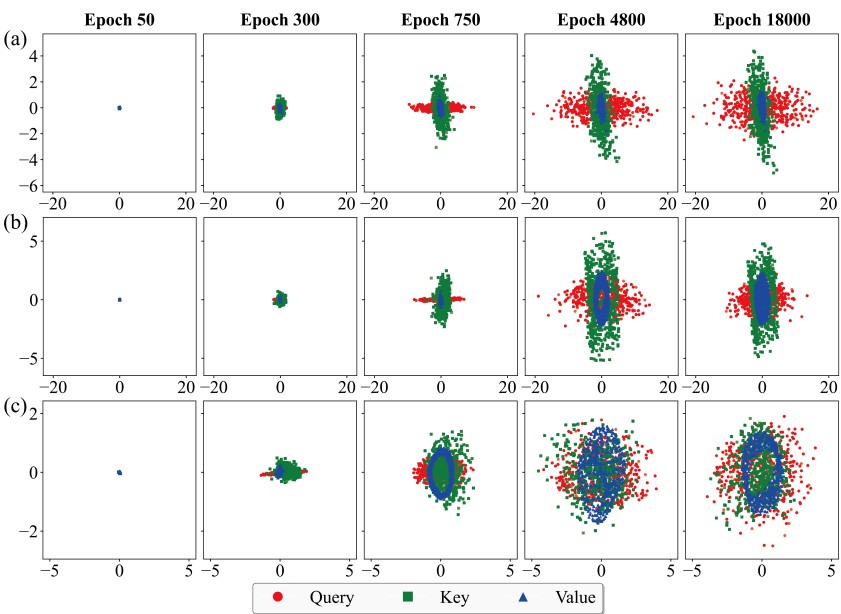

Figure 9: The results corresponding to operand $c$.

## C.2 VALUE VECTORS VISUALIZATION FOR OTHER TOKENS

In this section, we visualize the embedding positions of tokens 11~20 and 21~30. Consistent with the findings for tokens 1~10 presented in the main text, the results reveal distinct geometric patterns across learning phases:

- Memorization (Blue): Embeddings exhibit no discernible spatial organization and remain densely clustered;

- Semi-grokking (Orange): While no coherent structure emerges, embeddings become more dispersed;

- Grokking (Green): Merely 10 consecutive tokens form a well-defined circular arrangement with maximally separated embeddings for adjacent numerical tokens.

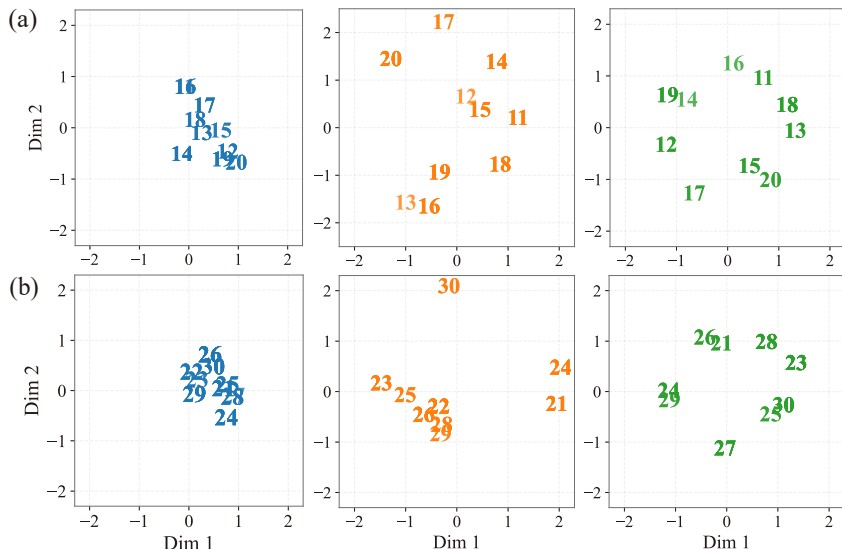

Figure 10: Embeddings of (a) token 11~20 and (b) 21~30 for operand $a$. Blue: memorization; Orange: semi-grokking; Green: grokking.

### C.3 VALUE VECTORS VISUALIZATION FOR OTHER OPERANDS

Figures11 (a) and (b) present the 2D embeddings of tokens 1~10 for operands $b$ and $c$ after PCA, respectively. The results for operands $b$ and $c$, are the same as operand $a$ in the main article.

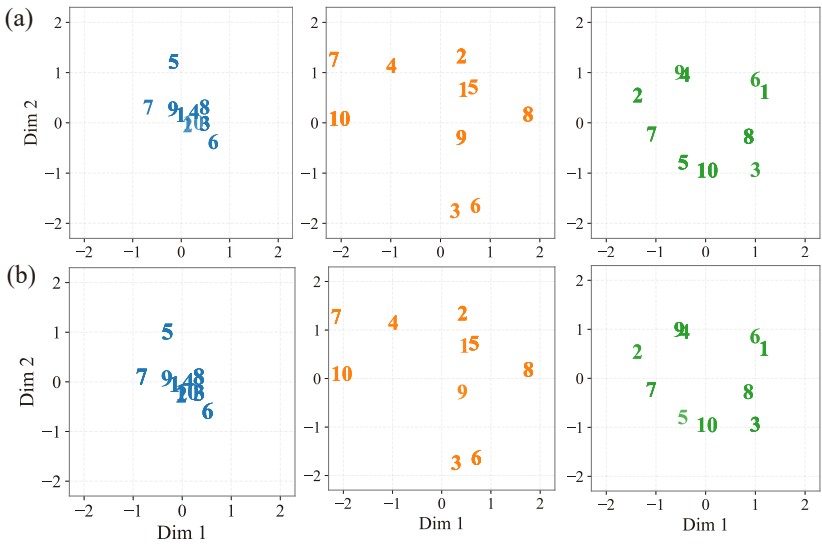

Figure 11: Embeddings of token 1~10 for (a) operand $b$ and (b) operand $c$. Blue: memorization; Orange: semi-grokking; Green: grokking.

# D  GROKKING ACCELERATION EXPERIMENT

## D.1  EXPERIMENT DETAILS

In each accelerating grokking experiment, we conduct 15 random seeds to report the results presented in the main text.

For the modular arithmetic tasks, we train the model for 18,000 epochs. And calculate the proportion of correct operand $c$. For the Tense-Inflection task, we train the model for 1,000,000 steps. Following previous studies (McCoy et al., 2020) (Murty et al., 2023), we measure the proportion of target verbs that are correctly inflected.

## D.2  R2G LOSS EXPERIMENT IN TENSE-INFLECTION TASK

Due to the differences among tasks, we adapt a reasonable variant of the R2G loss for the tense-inflection task. Since the model has 4 layers, we apply the R2G loss to the attention module of each layer and then take the average.

In addition, consistent with prior work, the tense-inflection dataset is generated from a vocabulary with a limited number of words. Since the shortest sentence length is 3, and with the sentence-final punctuation and the tense mark added, the total length becomes 5. Therefore, we select the first five token positions as the targets for R2G loss separation. This choice aims to cover as many tokens as possible while avoiding excessive duplication, thereby applying an appropriate strength of separation to different tokens.

Following the same methodology as in the main text, we randomly sample a pair of data instances, perform separation on the tokens at the first five corresponding positions for each pair. The algorithm pseudocode is shown in Algorithm 2.

---

**Algorithm 2** R2G Loss in Tense-Inflection Task

---

1: **def R2G_Loss_PerLayer**$(V)$
2:    *Input:* $V \in \mathbb{R}^{B \times L \times D}$ {value tensor of one layer}
3:    $K \leftarrow 5$ {first five token positions}
4:    **assert** $L \geq K$
5:    $perm \leftarrow \mathrm{RandomPermutation}(B)$
6:    $total\_distance \leftarrow 0$
7: **for** $j \leftarrow 0$ **to** $K - 1$ **do**
8:     $\mathbf{v} \leftarrow V[:, j, :]$
9:     $\mathbf{v}' \leftarrow V[perm, j, :]$
10:     $d \leftarrow \|\mathbf{v} - \mathbf{v}'\|_2$
11:     $total\_distance \leftarrow total\_distance + d$
12: **end for**
13:    **return** $total\_distance / K$
14:
15: **def R2G_Loss_AllLayers**$(\{V^{(1)}, V^{(2)}, V^{(3)}, V^{(4)}\})$
16:    *Input:* $V^{(\ell)}$ is the value tensor for layer $\ell$
17:    $loss \leftarrow 0$
18: **for** $\ell \leftarrow 1$ **to** $4$ **do**
19:     $loss \leftarrow loss + \mathrm{R2G\_Loss\_PerLayer}(V^{(\ell)})$
20: **end for**
21:    **return** $loss / 4$

---

## D.3  PHASE TRANSITION FACILITATED BY R2G LOSS

To further demonstrate the effect of R2G Loss in promoting the model's generalization capabilities, we present the results from 15 random repeat experiments, showing how our method pushes the training dynamics from one phase to another. Results of five tasks in the main text are shown in Figure 12. Although there are a few failures, the overall level of generalization improves, successfully elevating failed, memorization, or semi-grokking to a higher level of generalization.

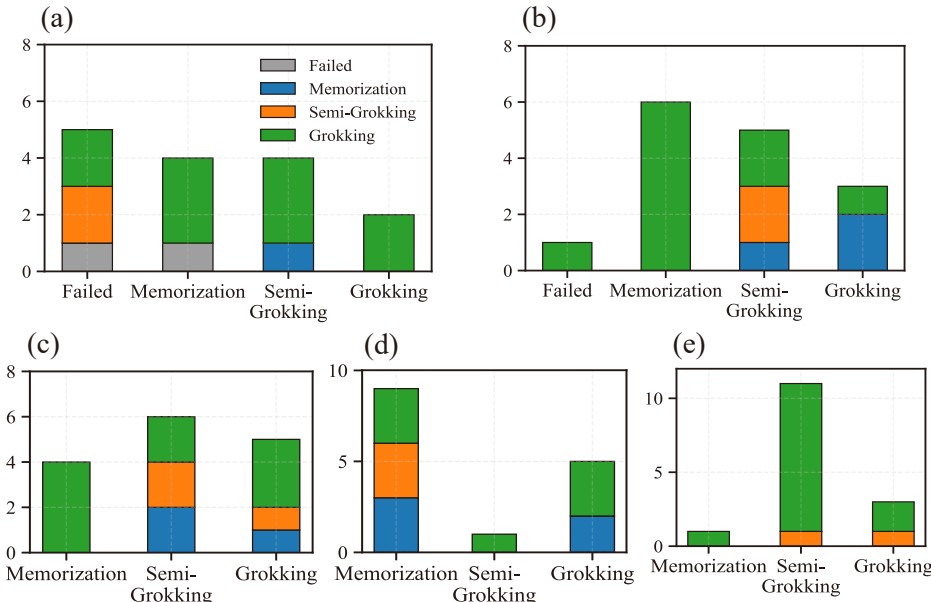

Figure 12: Phase Transition Facilitated by R2G Loss. The x-axis represents the original phases, while the y-axis represents the times of transition to each phase. (a) $P = 500$ Modular Addition Task. (b) $P = 499$ Modular Addition Task. (c) $P = 113$ Modular Addition Task. (d) $P = 113$ Modular Multiplication Task. (e) Tense-Inflection Task.

## E  THE USE OF LARGE LANGUAGE MODELS (LLMS)

During the writing of this paper, we used Large Language Models (LLMs) as auxiliary tools to polish and optimize the language and expression of certain paragraphs, particularly to enhance the academic writing style. All research design, experimental implementation, data analysis, and core academic content were independently completed by human authors. The LLMs only assisted in improving the clarity, coherence, and academic rigor of the writing. LLMs did not participate in formulating any scientific conclusions, nor did they contribute independently to the research direction or innovations. All content generated with LLM assistance has been thoroughly reviewed and edited by the authors, who take full responsibility for the accuracy, originality, and academic integrity of the paper.

