# OpenReview forum: "R2G Loss Accelerates Grokking in Transformer Models"
_ICLR.cc/2026/Conference — ICLR 2026 Conference Withdrawn Submission_

### Official Review · Reviewer_DdEK · 2025-10-30

**Soundness:** 1
**Presentation:** 1
**Contribution:** 2
**Rating:** 2
**Confidence:** 5

**Summary:**

This paper mechanistically observes that the V layer within the attention mechanism learns both the structural separation and the symbolic and numerical aspects of the data on the grokking tasks. With this finding, they develop the `R2G Loss`, which is the Euclidean distance between a layer's output projection and its randomly shuffled counterpart. They specifically use this loss on the `V` layer within the attention mechanism. This loss is weighted with $\alpha$ and added to the original loss. They observe that this loss can mitigate grokking, or in cases where a model would fail to learn the task can start to.

**Strengths:**

- The paper has a strong introduction where the related work is clear.

- It shows a potentially interesting method to mitigate grokking and improve model generalization.

- Highlights how mechanistic perspectives can be leveraged to improve model learning.

- I appreciate providing the source code and the reproducibility statement, and the effort to detail the exact training setup and conditions.

**Weaknesses:**

## Lack of Clarity/Inconsistent

### Use of Grokking
The terminology of grokking is often misused, which makes the paper hard to parse on the first read. Grokking within the literature refers to significantly delayed generalization [1]. Where the model memorizes the training data and then, significantly later, after little to no improvement on the test dataset, the model suddenly generalizes. This is referenced on line `034` with "The study of 'grokking” — a recently discovered phenomenon where neural networks abruptly transit from memorizing training data to achieving perfect generalization after prolonged training".  Therefore, I question if `accelerates` is the correct term; should it not be mitigates or reduces as this method attempts to avoid the significant delayed between memorization and generalization. Then later on in the paper line `147`, grokking is then redefined to mean `Grokking: Generalization dominates, yielding near-perfect validation accuracy`; this naturally causes a conflict with the previous definition; this term should be redefined as something like `Extreme Generalization`.

In line `075`, Figure 1 caption states: `R2G Loss training curves after smoothing show that different original training states are successfully promoted to achieve grokking`. For (a) and (b), it appears  that the `R2G Loss` completely mitigates grokking as the train and test accuracy increase in tandem, which is the opposite of promoting grokking. For (d) and (e), it appears that `R2G Loss` enables to model to better generalize with fewer training steps.

The stages: `Failed`, `Memorization`, `Semi-Grokking`, and `Grokking` are unclear. I think this would be better framed as `Failed`, `Memorization`, `Learning`, and `Extreme Generalization`. This would help make the paper clearer and reduce the chance of confusion for readers.

To add to this, the mix of definitions is very confusing when `R2G Loss` stands for `Repel to Grokking`, yet in parts of the paper, it is stated that it promotes grokking.

###  Notation

In the main body, $\lambda$ is used to represent weight decay, see line `182`, but in the appendix $\epsilon$ is used to represent weight decay, see line `665` .

## Figures

Subtitles on figures are unclear and uninformative. For example, Figure 1, the first figure is captioned `(a)`, which provides no information to the reader. This particular example could be improved to be `Baseline Performance`. Improving all sub-captions in this way will improve the clarity of the paper.

## Lack of Comparisons:

Although this method acknowledges methods such as Grokfast and Grokformer, there is no comparison. This would help improve the understanding of how effective this method is. For example, in a conventional setup as explored in GrokFast, their method makes the model generalize 50.49 times faster on $a\times b\, mod\, 97$. How does your method compare under similar conditions?

## Reproducibility:

Although the source code is provided, the documentation accompanying the codebase is lacking. This makes reproducing the results difficult. In addition, there is no code to produce the plots, which are central to the paper's claims.

## Lack of Ablations

Only the results of the best $\alpha$ value are reported. This can provide a distorted view of how effective this method is in practice. How exactly was this $\alpha$ value found for each task?  How brittle is the $\alpha$ hyper-parameter, and how does it affect learning? What is the hyperparameter landscape? Without knowing this, the impact of the work cannot be effectively assessed.

Given that transformers as the focus of the paper. I would expect that the `R2G Loss` be explored across all layer types, given it uses Euclidean distance and therefore can be. Nothing about the loss immediately suggests this would only be effective on the `V` layer. Exploring all layer types would help further the claim that it works for the reasons that say it works.  If only using the `R2G Loss` on the `V` layer results in an improvement, that would improve the argument behind only optimizing the `V` layer with the `RG2 Loss`.

On line `689` it states `There is a tendency that the more tokens need to be separated, the larger α is recommended.` However, for modular addition, where P=113, the $\alpha$ value is 0.08, but for modular multiplication, it is 0.01. This would suggest that the $\alpha$ value is more than due to the number of tokens.

**Questions:**

See questions in the weakness section but more concretely:

- Given this work positions itself as a method to eliminate grokking and does not go beyond this scope, can you compare this method to other methods such as GrokFast, GrokTransfer? Is this method better at mitigating grokking?

- What is the relationship of $\alpha$ and the corresponding affect on learning? How brittle is the $\alpha$ hyper-parameter, and how does it affect learning? What is the hyper parameter landscape?

- Figure 12 suggest this method appears to transition models that previously could generalize into models that either memorize or semi-grokk. Specifically Figure 12b the grokking phase now has Memorization and Grokking. What is the rational behind this occurring if this method is meant to mitigate grokking?

- Why is the only embedding position of token 1-10 of operand `a` shown in Figure 4 and not all tokens?

- What do the `Q` `K` and `V` 2d projections look like for the Tense-Inflection Task, how does this change using the `R2G Loss`.

---

### Official Review · Reviewer_P95t · 2025-10-31

**Soundness:** 2
**Presentation:** 2
**Contribution:** 1
**Rating:** 2
**Confidence:** 4

**Summary:**

This paper introduces R2G (Repel-to-Grokking) Loss, a simple regularization term that encourages geometric separation among attention value embeddings. The authors find that during grokking, transformer embeddings become more dispersed and argue that this represents a transition from symbolic memorization to numerical understanding. By enforcing this separation explicitly as a loss term, R2G Loss accelerates and amplifies grokking, improving grokking success from 13% to 67% on modular addition and from 20% to 87% on a tense-inflection language task.

**Strengths:**

- The proposed method is simple and requires no additional data or adjustments to model architecture.
- The authors motivate R2G with observations about internal transformer representations during training.
- The work is generally well-written and the experiments/figures are clear.

**Weaknesses:**

- The characterization of the memorization, semi-grokking, and grokking phases for modular arithmetic tasks through geometric properties of embeddings (while clearly presented) largely reproduces existing findings in the literature. For example, [1, 2, 3] show that transformer representations during grokking become roughly distributed evenly across a hypersphere, and [4, 5] argue that phase transitions between memorization and generalization can be formalized as distinct learning regimes. To strengthen the contribution, the authors could more explicitly clarify how their observations differ from these established results.
- The experimental results are relatively limited. The authors only evaluate their method on modular addition (for 3 values of P), modular multiplication (for 1 value of P), and a tense-inflection language task. To more convincingly demonstrate the strength of the method, a further evaluation on other grokking tasks (e.g. the ones in [1]) could be useful. See questions as well.

1. Liu et al., Omnigrok: Grokking Beyond Algorithmic Data, ICLR 2023.
2. Nanda et al., Progress measures for grokking via mechanistic interpretability, ICLR 2023.
3. Zhong et al., The Clock and the Pizza: Two Stories in Mechanistic Explanation of Neural Networks, NeurIPS 2023.
4. Varma et al., Explaining Grokking through Circuit Efficiency, 2023.
5. Huang et al., Unified View of Grokking, Double Descent and Emergent Abilities, COLM 2024.

**Questions:**

- In Appendix B.2 Table 1, the authors mention choosing a different value of the loss weight $\alpha$ for each of the five grokking subtasks. I'm wondering how consistent the grokking speedups were as $\alpha$ varied: did the authors observe them for almost all $\alpha$, and was there a trend observed? If the speedup is brittle with respect to $\alpha$, could it be possible the speedup is largely due to randomness of the optimization procedure?
- Can the authors explain how their method relates to Grokfast [1] or Hessian regularization [2]? The motivations are quite different from this work, but both methods produce similarly large speedups on modular arithmetic by adjusting the gradients used during training. Like [1, 2], can I intuitively understand R2G as effectively reducing the noise in the gradient signal by adding the additional contrastive loss?

1. Lee et al., Grokfast: Accelerated Grokking by Amplifying Slow Gradients.
2. Zhang et al., A Hessian View of Grokking in Mathematical Reasoning.

---

### Official Review · Reviewer_w2tP · 2025-11-01

**Soundness:** 2
**Presentation:** 2
**Contribution:** 2
**Rating:** 2
**Confidence:** 3

**Summary:**

This paper proposes the R2G loss, a method whose main idea is to encourage a more uniform representation in the embedding space. Empirical studies on different arithmetic tasks demonstrate that R2G can improve the model's performance.

**Strengths:**

1. This paper provides a mechanistic interpretation for grokking from the model's internal representation perspective.

2. This paper proposes the R2G loss, which is demonstrated to be effective in improving the model's performance.

**Weaknesses:**

1. The experimental statements are not entirely clear. For example, the terms "Memorization," "Semi-grokking," and "Grokking" lack a clear definition. It is unclear whether these phases are classified based on the training dataset size or the model's performance metrics. Furthermore, the results presented in Figures 3 and 4 appear insufficient to conclusively demonstrate that grokking correlates with the distance between embedding vectors; the evidence seems more correlative than causative.

2. The proposed loss function lacks validation on larger, more realistic datasets, which limits the assessment of its practical utility and generalizability beyond controlled algorithmic tasks.

3. The R2G Loss, as described in Algorithm 1, operates by increasing the distance between embeddings from different data samples. The paper would be strengthened by including ablation studies or comparisons with alternative regularization approaches. For instance, how does it compare to regularizing the distance between embeddings and a fixed random vector, or to other methods that encourage uniform distribution?

**Questions:**

Please refer to the weaknesses section above. If there are any misunderstandings on my part, please point them out, and I will reconsider my evaluation of this work.

---

### Official Review · Reviewer_d11T · 2025-11-06

**Soundness:** 2
**Presentation:** 1
**Contribution:** 2
**Rating:** 2
**Confidence:** 4

**Summary:**

The paper studies grokking through the lens of internal attention representations and proposes R2G (Repel-to-Grokking) Loss, which is claimed to “enforce structural separation” among attention V embeddings to accelerate grokking. Empirically, on modular arithmetic (addition/multiplication, various moduli) and a tense-inflection toy NLP task, the method reportedly increases the fraction of runs that reach the “grokking” phase (e.g., for P=500 modular addition, from 13% to 67% at fixed data size) and reduces “failed/memorization/semi-grokking” outcomes. The model studied is mainly a 1-layer decoder-only transformer (d=48, 4 heads). The paper visualizes Q/K/V embeddings (via PCA) and presents histograms of 48-D distances, arguing that in the grokking phase V-embeddings display a “circular/hyperspherical” separation and that adjacent digits become more separated than random pairs.

**Strengths:**

Simple, lightweight method. R2G is easy to bolt onto a standard transformer: it adds a distance term over shuffled V-embeddings, requires no extra modules or data, and operates directly on intermediate representations (Algorithm 1). This simplicity makes it broadly adaptable.

**Weaknesses:**

1.	Core loss definition vs. claimed effect.
The method is said to enlarge distances between randomly paired V embeddings, but the loss is defined as
($L = L_{\text{orig}} + \alpha \cdot |V - V'|_2$) (Algorithm 1 sums (|V_i - V'_i|^2)), with positive α values reported for all tasks. Minimizing this objective reduces distances, not enlarges them. This contradicts both the stated intuition (“repel”) and the narrative throughout the paper. Please clarify whether the implementation actually maximizes the distance term (e.g., by using a negative coefficient or reversing the sign), or if there is a typographical error. As written, the math undermines the central claim.

2.	Motivation ordering / missing bridge to Sec. 3.3.
Section 3.3 introduces R2G before the paper clearly motivates why distance between embeddings should matter for grokking. The explicit research question—“Does model grokking correlate with distance between embedding vectors?”—appears later in Section 4, after PCA plots. That makes 3.3 feel like it “comes from nowhere.” Restructuring to first pose and analyze the correlation question, and only then derive the loss, would make the narrative coherent.

3.	Correlation vs. causation not established.
The paper asserts that grokking “strongly correlates” with V-space structure and uses that to justify R2G, but evidence is descriptive (PCA snapshots; histogram shifts). There are no quantitative correlation measures across training (e.g., epoch-wise correlation between mean pairwise distances and validation accuracy) nor controlled interventions that separate representation dispersion from other effects (e.g., embedding norm growth).

4.	Baselines are underspecified / incomplete for a loss paper.
If the contribution is a representation-repulsion loss, comparisons should include standard dispersion/contrastive objectives (e.g., cosine/orthogonality regularizers, hyperspherical energy, supervised contrastive/InfoNCE-style terms) and stronger “delay-reduction” baselines beyond the few cited thematically. Without this, it’s unclear whether R2G is better than well-known alternatives that also separate embeddings.

5.	Risk that the loss just inflates norms.
Euclidean distance can be increased by growing vector norms rather than inducing meaningful geometry. There is no analysis of embedding norm distributions, cosine distances, or explicit norm controls. This weakens the mechanistic claim that R2G encourages “structural separation.” (A cosine-based term or norm-normalization ablation would help.)


6.	Generality claims exceed evidence.
Results are on small toy setups—a 1-layer, d=48 model and a synthetic tense-inflection task with custom loss application to the first five token positions. It’s unclear whether the method helps modern multi-layer transformers on realistic language or vision tasks without hand-selecting token positions.

7.	Hyperparameter sensitivity and compute budget.
α varies per task with no sensitivity analysis or schedule; training goes to 18k epochs for arithmetic (unusually long), yet compute/time isn’t discussed. The method’s stability region and overhead are unknown.

8.	Clarity issues.
The paper doesn’t clearly explain how “distance between embedding vectors affects grokking.” Mechanism is asserted, not derived. Several sections read like preliminaries without tightening the causal story, and Section 3.3 indeed feels abrupt relative to the narrative setup.

**Questions:**

1.	Sign / implementation of R2G: Are you maximizing (|V - V'|) or minimizing it? With ($L=L_{\text{orig}}+\alpha|V-V'|$) and positive α (Table 1), SGD will shrink distances. Please reconcile this with both Algorithm 1 and the “repel” intuition.
2.	Did you compute epoch-wise correlations between average pairwise V-distances (or hyperspherical energy) and validation accuracy to substantiate “strong correlation”? If yes, please report coefficients and CIs.
3.	Why apply R2G only to V? What happens if applied to Q/K or all three?
4.	Is the gain due to norm inflation? Please report embedding norm distributions and cosine distance analyses; consider norm-normalizing V before computing the loss.
5.	Baselines: how does R2G compare against supervised contrastive, orthogonality constraints, or hyperspherical regularizers under equal compute?
6.	For tense inflection, R2G is applied to the first five positions only. How sensitive are results to this design choice and to sentence length?
7.	Provide ablation on α (grid/schedule) and report sensitivity; any signs of instability or training collapse at larger α?
8.	Can you show that the claimed 2-D “circle” corresponds to a meaningful high-D structure beyond PCA (e.g., hyperspherical uniformity tests, RIP/packing metrics)?
9.	How do results scale to deeper/wider transformers and to real-world tasks (e.g., standard text classification, algorithmic extrapolation beyond modular arithmetic), without bespoke positional selection?

---

### Note · Authors · 2025-12-02

**Comment:**

Dear Area Chair and Reviewers,

Thank you for your constructive feedback and detailed comments. After carefully considering the reviews, we have decided to withdraw our submission from ICLR 2026. We appreciate the time and effort you have dedicated to reviewing our work.

Best regards,
The Authors

**Withdrawal Confirmation:**

I have read and agree with the venue's withdrawal policy on behalf of myself and my co-authors.